# Unraveling the Effect of Strain Rate and Temperature on the Heterogeneous Mechanical Behavior of Polymer Nanocomposites via Atomistic Simulations and Continuum Models

**DOI:** 10.3390/polym16172530

**Published:** 2024-09-06

**Authors:** Ali A. Youssef, Hilal Reda, Vagelis Harmandaris

**Affiliations:** 1Computation-Based Science and Technology Research Center, The Cyprus Institute, 2121 Nicosia, Cyprus; a.a.youssef@cyi.ac.cy (A.A.Y.); h.reda@cyi.ac.cy (H.R.); 2Department of Applied Mathematics, University of Crete, GR-71409 Heraklion, Greece; 3Institute of Applied and Computational Mathematics, Foundation for Research and Technology-Hellas, IACM/FORTH, GR-71110 Heraklion, Greece

**Keywords:** polymer nanocomposites, atomistic simulations, heterogeneous mechanical properties, temperature and strain rate effect

## Abstract

Polymer nanocomposites are characterized by heterogeneous mechanical behavior and performance, which is mainly controlled by the interaction between the nanofiller and the polymer matrix. Optimizing their material performance in engineering applications requires understanding how both the temperature and strain rate of the applied deformation affect mechanical properties. This work investigates the effect of strain rate and temperature on the mechanical properties of poly(ethylene oxide)/silica (PEO/SiO_2_) nanocomposites, revealing their behavior in both the melt and glassy states, via atomistic molecular dynamics simulations and continuum models. In the glassy state, the results indicate that Young’s modulus increases by up to 99.7% as the strain rate rises from 1.0 × 10^−7^ fs^−1^ to 1.0 × 10^−4^ fs^−1^, while Poisson’s ratio decreases by up to 39.8% over the same range. These effects become even more pronounced in the melt state. Conversely, higher temperatures lead to an opposing trend. A local, per-atom analysis of stress and strain fields reveals broader variability in the local strain of the PEO/SiO_2_ nanocomposites as temperature increases and/or the deformation rate decreases. Both interphase and matrix regions lose rigidity at higher temperatures and lower strain rates, blurring their distinctiveness. The results of the atomistic simulations concerning the elastic modulus and Poisson’s ratio are in good agreement with the predictions of the Richeton–Ji model. Additionally, these findings can be leveraged to design advanced polymer composites with tailored mechanical properties and could optimize structural components by enhancing their performance under diverse engineering conditions.

## 1. Introduction

Neat polymers are of significant importance in several technical applications; yet their comparatively low stiffness and strength pose a distinct constraint in such contexts. To expand the scope of their use, various reinforcements are used to enhance the thermal, physical, and mechanical characteristics of these polymers [1,2,3,4,5,6]. Over the past few years, there has been a significant surge in fascination with polymer nanocomposite (PNC) materials, largely due to their exceptional mechanical capabilities and extensive range of potential applications. In particular, poly(ethylene oxide)/silica, PEO/SiO_2_, nanocomposites have shown great promise in the fields of engineering and (bio)nanotechnology, attracting much attention for ongoing research and development efforts [7,8,9,10,11]. One of the characteristics of PEO/SiO_2_ nanocomposites is the relatively strong attractive interactions between polymer chains and the nanofillers, due to hydrogen bonding, which result in strongly adsorbed polymer chains on the surfaces of the silica nanoparticles. The adsorption process plays a critical role in the stabilization of many systems, thus giving substantial potential for usage across a wide range of applications in several industrial fields, including, among others, energy and medicine [12,13,14,15,16,17,18,19,20]. Consequently, it is imperative to conduct a comprehensive examination of the mechanical characteristics of polymer nanocomposites, such as PEO/SiO_2_, to determine the optimal material for certain applications.

Computational modeling and molecular simulations are invaluable tools, complementary to experiments, for investigating the mechanical properties of materials and their nanocomposites [21,22,23,24]. Simulations provide significant information on the deformation, stress resistance, and mechanical characteristics of materials under various conditions [25]. Through the use of atomistic simulations in particular, one can obtain a more comprehensive understanding of the underlying mechanisms that govern the mechanical properties of polymeric materials containing nanostructured components at the atomic scale. Understanding this information allows for the creation and design of cutting-edge materials with enhanced mechanical properties, broadening their range of possible uses. Simulations have facilitated extensive research, enabling a thorough exploration of the structural and dynamic properties specific to PEO/SiO_2_ systems [26,27,28,29,30].

Polymer nanocomposites display heterogeneous mechanical properties as a result of intricate interactions between the polymer matrix and nanofillers. As a result of its inherent heterogeneity, predicting mechanical properties based on the molecular structure is a challenging task. Temperature fluctuations can modify bonding and movement at the interface, whereas varying strain rates impact the distribution of stress and the mechanisms of deformation. Therefore, the examination and understanding of the synergistic impacts of these factors on nanostructured systems based on polymers is crucial to optimize their functionality and broaden their potential applications across diverse industries. In this context, it becomes essential to explore the deformation behavior of polymer nanocomposites at various strain rates and temperatures, as it is well known that the deformation rate and the temperature have a significant impact on the mechanical characteristics of polymer nanocomposites [3,21,31,32,33,34]. The complex character of the behavior of polymer nanocomposites under different environmental circumstances is shown by this ever-changing relationship. The important and significant individual impacts of strain rate and temperature on the mechanical properties of polymer-based nanostructured systems are widely recognized in the academic literature [35,36,37,38,39,40]. For example, it has been shown that the mechanical properties of polymer-based nanostructured systems are significantly impacted by strain rate [3,35,36,37,38]; the strain rate directly influences the rate of deformation, defining the material’s responsiveness to external loads and ability to endure various loading circumstances. Furthermore, it has a substantial impact on effective mechanical behavior, especially during the elastic–plastic transition [41,42,43]. The response of polymer-based materials to external loads, influenced by the deformation rate, is affected by the interplay between the relaxation time of the polymeric atoms or segments and the distinctive ‘deformation time’ (inverse deformation rate) induced by the externally applied load, as these materials possess a wide range of relaxation durations [11,33,44,45]. According to previous studies, there is a non-linear correlation between the initial Young’s modulus and the strain rate, similar to the behavior observed in yield stress under high strain rates. Previous studies have underscored the interrelation between yield and segmental mobility associated with the β-relaxation process. This link extends to the yield stress, since the elevation of the initial Young’s modulus with decreasing temperature is attributed to the decrease in the β-relaxation (enormous increase in segmental relaxation times) of polymer chains [32,46,47,48,49,50,51]. Therefore, it is imperative to evaluate the influence of the strain rate on mechanical properties when designing structures that withstand substantial strain rates without premature or unexpected failure [39,40,41,42,43,52,53]. Variations in temperature induce changes in the mechanical and physical characteristics of polymers, regardless of the presence of reinforcing particles. Furthermore, temperature plays a fundamental role in altering the thermal energy and molecular motion of the material, affecting its stiffness, strength, and general mechanical behavior [30,54]. The alteration is the result of the viscoelastic characteristics of the polymers, leading to phenomena such as creep, relaxation, and susceptibility to high temperatures [55,56,57,58,59,60,61].

Finite element simulations have shown that the Young’s modulus of polymer nanocomposites decreases with increasing temperature, as reported in previous studies [62]. The molecular structural mechanics method also revealed that when the temperature increased, the elastic rigidity of the polymer nanocomposites decreased significantly [63]. Drozdov [64] proposed a relationship for the temperature-dependent elastic modulus; however, it is only applicable below the glass transition temperature. Although some models have been developed in the literature to predict elastic properties in the elastic region [65], their applicability is restricted to temperatures that exceed the glass transition temperature. To fill this void and encompass the complete range of temperatures both below and above $T_{g}$, Mahieux and Reifsnider [66] constructed a statistical model that analyzes the rigidity of polymers at a variety of temperatures.

In light of the previous information, it is imperative to develop a constitutive model that takes into account the aforementioned factors [3,67]. Expanding on this notion, Richeton et al. [32] present a methodology that integrates the influences of strain rate (or frequency) and temperature, although it is limited to pure polymers. An extension of this model, usually denoted as the Richeton–Ji (RJ) model, can be used to predict the Young’s modulus of nanocomposite materials [32,34]. This model is employed to predict the mechanical properties of the polymer nanocomposite, including Young’s modulus and Poisson’s ratio, thereby capturing the effect of SiO_2_ nanofillers on the polymer matrix and providing a framework for understanding how these fillers influence the composite material’s overall mechanical behavior. Moreover, there are other models frequently utilized to estimate the elastic performance of polymer nanocomposites. Within this spectrum, the Halpin–Tsai model, meticulously crafted by Halpin and Kardos [68] to determine the elastic modulus of composite materials, is particularly noteworthy and is garnering attention. Further enriching our understanding is the Tandon–Weng (TW) model [69], which intricately weaves together the theoretical foundations of Mori and Tanaka [70] to predict the elastic modulus of nanocomposites. Furthermore, the Richeton–Tandon–Weng (RTW) model [3] combines the Richeton and TW models, thus improving analytical capabilities in this field [21]. In this study, to deepen our understanding of PEO/SiO_2_ nanocomposites, the Richeton–Ji (RJ) model [32,33,71] was used to forecast the elastic modulus of the nanocomposites [3,21]. When the results obtained from this model are juxtaposed with those of the MD simulations, the aim is to obtain a thorough understanding of the nanocomposite system under scrutiny.

While previous studies have explored the mechanical properties of polymer nanocomposites, there remains a lack of comprehensive understanding regarding the synergistic effects of strain rate and temperature on the heterogeneous mechanical behavior of PEO/SiO_2_ nanocomposites. Most existing research has focused on isolated factors, either strain rate or temperature, without thoroughly investigating their combined impact on the material’s performance at the atomic scale. This study addresses this gap by providing a detailed analysis of how these two critical parameters interact to influence the deformation behavior of PEO/SiO_2_ nanocomposites. Through atomistic simulations complemented by continuum models, this research offers new insights into the temperature- and strain-rate-dependent mechanical properties of these nanocomposites, particularly their local stress and strain distributions. The findings contribute to optimizing the design and application of PEO/SiO_2_ nanocomposites in various engineering fields where precise control over mechanical behavior under different environmental conditions is crucial.

The present paper is organized as follows: The next section provides an in-depth analysis of the simulation technique employed, the model used, and the intricate details pertaining to the simulated system. Section 3 of the manuscript discusses and examines the results derived from the MD simulation, together with the results utilized from the Richeton–Ji (RJ) model to estimate the elastic modulus of the nanocomposites. Section 4 discusses the mobility of polymer chains through the analysis of mean-squared displacement (MSD) computations. Finally, in Section 5, the conclusion is presented, which elucidates the key findings and contributions.

## 2. Models and Methods

### 2.1. Preparation and Equilibration of Model Configurations

Molecular dynamics (MD) simulations are widely employed in scientific research to simulate the temporal behavior of a collection of atoms that interact with each other. This work involves performing atomistic MD simulations on a model system consisting of PEO/SiO_2_ nanocomposites. The nanocomposite PEO/SiO_2_ model is made up of 33 wt% (12.7 vol%) silica nanoparticles. The silica nanoparticles possess an amorphous structure and are roughly spherical in shape, with an estimated diameter of approximately 4 nm. The nanocomposite contains 48 PEO chains, which consist of 50 monomers each. The molecular weight of the PEO chains is around 2.2 kDa. Additional information on the molecular structure of the PEO and silica nanoparticles can be found in previous works [11,29,72,73]. The force field parameters for the PEO and silica can be found in Appendix A. Table A1 provides a comprehensive overview of the non-bonded interactions, whereas Table A2 delineates the bonded interactions. The Lorentz–Berthelot mixing rule was employed to compute the non-bonded interactions between atoms of PEO and SiO_2_.

This study examines the mechanical response of polymeric nanocomposites (PNCs) under various tensile deformations. Deformations were applied at different strain rates, including 1.0 × 10^−4^, 5.0 × 10^−5^, 1.0 × 10^−5^, 5.0 × 10^−6^, 1.0 × 10^−6^, 5.0 × 10^−7^, and 1.0 × 10^−7^ fs^−1^, on three systems, maintaining constant temperatures equal to 220 K (glassy state), 270 K (close to the glass transition temperature, $T_{g}$), and 330 K (melt state) each. It is worth mentioning that the $T_{g}$ value of the model PEO/SiO_2_ systems was previously ascertained to be around 270 K in our previous studies [11,29]. Furthermore, tensile deformations were performed under different thermal conditions, spanning temperatures from 150 K to 400 K, using two different strain rates, 1.0 × 10^−5^ fs^−1^ and 1.0 × 10^−6^ fs^−1^. Table 1 presents the temperature and strain rate values and their corresponding deformation times, calculated as the inverse of the strain rate. The “deformation time”, $t_{D}$, (inverse of strain rate) is of significant importance in describing the material’s behavior, in relation to its relaxation times. The time step in all simulations was 1 fs.

To investigate the effect of strain rate while maintaining a constant temperature, the initial PEO/SiO_2_ model systems (data files that include molecular configuration and specifications, including simulation box dimensions, atom types, atom coordinates, and force field parameters) were utilized. To ensure accurate results, several (here 10) uncorrelated configurations were created by performing long equilibrium MD simulations at a high temperature of 400 K. Then, each configuration was cooled down to the desired temperature, with a cooling rate of 10 K/ns, followed by additional MD simulations of a few ns to achieve thermal and small-scale equilibration.

After equilibration, a controlled deformation along the x-direction is performed for each system, utilizing the ${NP}_{YY}P_{ZZ}T$ ensemble, wherein a constant temperature is maintained alongside normal stresses applied in the y and z directions. This is achieved through the use of a Nose–Hoover thermostat for temperature control and a Parrinello–Rahman barostat for stress imposition. In this work, deformation tests were applied up to a maximum global strain value of 0.4. However, the analysis focuses on the linear (low strain value) regime. To maintain the integrity of the systems, periodic boundary conditions were consistently applied in all three directions throughout the approach.

### 2.2. Deformation Process

Due to the isotropic nature exhibited by the PEO/SiO_2_ PNCs in light of the nearly spherical shape of the nanofiller, a single applied axial tensile deformation test (in this case, in the x-direction) is adequate to determine their comprehensive mechanical properties. Therefore, it is possible to deduce the rigidity matrix by determining the numerical values of the Young’s modulus, denoted as E, and the Poisson’s ratio, denoted as ν. Snapshots of the PEO/SiO_2_ model systems before and after tensile deformation (for the 0.4 strain value) are shown in Figure 1.

The current investigation involved the implementation of the deformation process by performing MD simulations using LAMMPS [74]. The mechanical parameters of the model systems, including the Young’s modulus and Poisson’s ratio, are determined by analyzing the stress vs. strain curves. In addition, local stress values are computed using the atomic virial formalism. This version of the stress tensor corresponds to Harasima’s concept of the local stress tensor, which states that the stress caused by the interaction of two particles is spread equally throughout the regions where the particles exist. It should also be noted that the proposed method can be applied if Hardy stresses are used [75]. Here, local stress profiles are determined by calculating the mean PEO the sum of atom stresses in the regions that correspond to the interphase and matrix regions around the silica nanoparticle.

To further enrich the investigation, the study implemented a new method to directly examine local deformation fields at the atomic level [75,76]. This approach allows us to calculate the deformation gradient for every atom using an intricate minimization problem. Valuable information about the distribution of local strain fields in the atomistic model was obtained by analyzing the positions of the atom of interest and its neighboring atoms within the specified cutoff radius ($r_{cut}$). The Lagrange Green strain tensor, which is defined in relation to the reference coordinates, played a vital role in the evaluation and analysis of these localized strain fields. More information about the calculation of the local, per-atom-defined deformation (strain) can be found in our previous works [11,77].

## 3. Results and Discussion

### 3.1. Macroscopic Mechanical Behavior of PEO/SiO_2_ Nanocomposites

The investigation of the model PEO/SiO_2_ systems began with a direct examination of their overall mechanical behavior under external deformation. As elucidated earlier, for our particular investigation, the entire rigidity matrix can be inferred from a single axial tensile test. This test procedure allows precise determination of both Young’s modulus, E, which measures the resistance to deformation under tensile loading, and Poisson’s ratio, ν, which measures the transverse deformation response in relation to axial deformation.

Based on information on the relationship between overall stress and externally applied strain for all PEO/SiO_2_ systems, as illustrated in Appendix B, it is apparent that stress–strain measurements are significantly affected by both the applied strain rate and the temperature conditions within the system being studied. In particular, as the applied strain rate increases, the PEO/SiO_2_ systems exhibit higher maximum stresses for a given magnitude of strain. On the contrary, temperature impacts exhibit a contrasting pattern, whereby higher temperatures are associated with lower peak stresses in the system. Figure 2 presents a comprehensive three-dimensional depiction of how the combination of strain rate and temperature affects both the Young’s modulus and Poisson’s ratio in the studied system. The Young’s modulus is obtained by determining the slope of the elastic region (linear regime) of the stress–strain curve within the low-strain (ɛ<0.05) values. Within this domain, stress exhibits a linear dependence on strain, facilitating precise characterization of the Young’s modulus, whereas the Poisson’s ratio represents the slope of the longitudinal strain-transverse strain curve. Figure 2 clearly demonstrates that the combined influence of strain rate and temperature has a substantial impact on the behavior of the system. In Figure 2a, whenever the system is subjected to a high strain rate and low temperature, it shows a high Young’s modulus, indicating the characteristics of a rigid structure and the brittle behavior of the material. If the system is subjected to deformation at an extremely slow rate and at a temperature significantly higher than the glass transition temperature, the system will exhibit much softer behavior compared to its prior condition. The results of the Poisson’s ratio in Figure 2b reflect the same conclusion, where lower values of this ratio indicate a higher level of system hardness and vice versa.

The comprehensive three-dimensional analysis presented in Figure 2 demonstrates the interplay between strain rate and temperature on the mechanical properties of the nanocomposites. The observed trends in Young’s modulus and Poisson’s ratio, as shown in the figure, are consistent with theoretical expectations, where a high strain rate corresponds to increased material stiffness, and a high temperature results in reduced rigidity. These observations are supported by the fundamental principles of polymer mechanics, which suggest that strain rate and temperature influence the molecular dynamics and, consequently, the macroscopic mechanical properties of the material.

To enhance clarity, a focused analysis was performed on the changes in the Young’s modulus and Poisson’s ratio with respect to the strain rate across several temperature ranges: below, at, and above the glassy transition point, as described in Section 3.1.1. Furthermore, in Section 3.1.2, the dependence of the Young’s modulus and Poisson’s ratio on temperature were analyzed using consistent and distinct strain rate values. In a 2D analysis, complexities are reduced, making it easier to apply theoretical/analytical frameworks and equations derived from previous research [32,33].

#### 3.1.1. Dependence of the Linear Mechanical Properties on Strain Rate

Figure 3 shows the findings related to the Young’s modulus (E) and Poisson’s ratio (ν) in relation to the strain rate ($\dot{\varepsilon }$) under various temperature conditions. These conditions encompass temperatures that are below, equal to, and above the glassy transition temperature ($T_{g}$). Furthermore, Figure 3a illustrates the results derived from the theoretical Richeton–Ji (RJ) model, depicting the dependence of the elastic Young’s modulus on both the temperature and the deformation rate. Note that the strain rate axis is graphed using a logarithmic scale. The equation of the Richeton–Ji model incorporates parameters associated with the temperature (*T*), strain rate ($\dot{\varepsilon }$), and material constants. According to this model, the temperature and strain rate dependence of Young’s modulus, *E*, and Poisson’s ratio, ν, follow a modified exponential form [21]:(1)$$E\left(T,\dot{\varepsilon }\right)=E_{c}\left(\dot{\varepsilon }\right)*exp\left(-{\left(\frac{T}{T_{g}\left(\dot{\varepsilon }\right)}\right)}^{m}\right),\;ʋ\left(T,\dot{\varepsilon }\right)={ʋ}_{c}\left(\dot{\varepsilon }\right)*exp\left(-{\left(\frac{T}{T_{g}\left(\dot{\varepsilon }\right)}\right)}^{m}\right),$$
where *m* represents the Weibull modulus, which characterizes the activation energy for bond breakage in connection with temperature. $E_{c}$ is the composite elastic modulus (representing the overall stiffness) of the nanocomposite. It accounts for the influence of nanofillers by incorporating their volume fraction, the stiffness ratio between the polymer and the nanofillers, and the Young’s modulus of the nanofillers. For a detailed explanation of the step-by-step computation using the equation above, please refer to Appendix C. This approach allows the model to accurately predict the nanocomposite’s elastic modulus, taking into account the effects of nanofillers across different temperatures and strain rates. For further details on the formulation, please refer to reference [21]. As shown in Figure 3a, it is evident that Young’s modulus aligns closely with the predictions of the RJ model in the range of strain rates for systems at particular temperature values. The same model was used to compare the Poisson’s ratio results with those obtained from MD simulations (Figure 3b). The model closely corresponds to the results of the MD simulations, affirming the consistency between the model and the MD technique. As shown in Table 2, based on global results, the Weibull modulus values, derived from fitting the MD simulation curves using the Young’s modulus formulation of the RJ model, range between 1.8 and 7.4 for systems under various temperature conditions, showing an increase as the system approaches a molten state. However, a contrasting pattern was noted when employing the Poisson’s ratio formulation, resulting in a reduction in the Weibull modulus from 5.91 to −4.91 as the system’s temperature increased from 220 K to 330 K. The data presented in Table 2 clearly demonstrate that reducing the strain rate applied to the system has an identical impact on the Weibull modulus as raising the temperature of the system.

Based on the data shown in Figure 3a, it becomes clear that an increase in the strain rate leads to an increase in the elastic modulus. As an example, for systems at temperature 220 K, the increase in strain rate from 1.0 × 10^−7^ fs^−1^ to 5.0 × 10^−7^ fs^−1^ yields a rise in the Young’s modulus by approximately 9.2%. The percentage increase becomes significantly higher, reaching up to 99.7%, from 1.0 × 10^−7^ fs^−1^ to 1.0 × 10^−4^ fs^−1^. This escalation is particularly pronounced at strain rates greater than 1.0 × 10^−5^ fs^−1^. This happens because at higher strain rates, which correspond to low deformation times, the relaxation and reorientation of polymer chains are impeded, leading to an increase in material orderliness and stiffness. This is consistent with a previous study that revealed that increasing the strain rate hinders chain relaxation, thus reducing its molecular mobility [78,79]. On the other hand, as the temperature of the system increases, the improvements in the Young’s modulus with the increasing strain rate become more pronounced. For the system at 270 K, the percentage increase in Young’s modulus reaches up to 214.8% at the highest applied strain rate (1.0 × 10^−4^ fs^−1^), while at 330 K, the improvement is even more substantial. These values indicate that the influence of strain rate becomes more apparent in the melt state, demonstrating that higher temperatures enhance the sensitivity of the modulus to strain rate variations. On the contrary, the temperature of the system has a greater impact on Young’s modulus at strain rates smaller than 1.0 × 10^−5^ fs^−1^. In such circumstances, the Young’s modulus decreases with temperature. When the temperature is higher than the glass transition temperature ($T_{g}$), this impact is clearly noticeable. The Young’s modulus is observed to decrease significantly for lower strain rate values (<1.0 × 10^−5^ fs^−1^) at temperatures below $T_{g}$ compared to earlier states. The opposite of the aforementioned situation is shown in Figure 3b, where a rise in strain rate causes a reduction in Poisson’s ratio, indicating properties similar to those of a rigid system. As the strain rate increases, this behavior becomes more noticeable. For the system at 220 K, the Poisson’s ratio decreases by up to 39.8% as the strain rate increases from 1.0 × 10^−7^ fs^−1^ to 1.0 × 10^−4^ fs^−1^. At 270 K, the reduction reaches 43.9% over the same range of strain rates. In contrast, at 330 K, the decrease is 42.1%. This indicates that while the Poisson’s ratio consistently decreases with increasing strain rate at all temperatures, the magnitude of the reduction becomes more substantial at higher temperatures, suggesting that the material’s response to the strain rate is more pronounced as it approaches its melt state. In particular, when the strain rates exceed 1.0 × 10^−5^ fs^−1^, the Poisson’s ratio demonstrates negligible susceptibility to changes in temperature. In contrast, the impact of temperature becomes more noticeable with lower deformation rates and for temperatures that exceed the glass transition temperature ($T_{g}$).

#### 3.1.2. Dependence of the Linear Mechanical Properties on Temperature

According to the stress–strain curves presented in Appendix B, as the strain rate decreases from 1.0 × 10^−5^ fs^−1^ to 1.0 × 10^−6^ fs^−1^, a considerable disparity in the stress–strain curves becomes visible. Data indicate that the system experiences a reduction in both rigidity and strength when subjected to low strain rates and elevated temperatures, in contrast to scenarios with similar temperatures but higher strain rates, where rigidity is preserved. As expected, a gradual increase in peak strength and rigidity was found in the latter circumstances.

Based on the information provided in Figure 4a, it is feasible to draw the conclusion that a slight change in the Young’s modulus was observed within the temperature range of 150 K to nearly 270 K (which is equivalent to $T_{g}$, the glass transition temperature). This is due to the restricted molecular mobility of the polymer chains as a result of the low thermal expansion and stiffer material below the glass transition temperature ($T_{g}$). However, as the temperature approaches this threshold, the Young’s modulus begins to decrease in a detectable manner, particularly when the material is subjected to low strain rates. For example, at a strain rate of 1.0 × 10^−5^ fs^−1^, the percentage decrease in the Young’s modulus increases from 6.69% to 64.79% as the temperature rises from 220 K to 400 K, compared to the system at 150 K. Similarly, at a strain rate of 1.0 × 10^−6^ fs^−1^, the percentage decrease reaches up to 97.24% at 400 K. This trend shows that the effect of temperature on the material’s modulus becomes more pronounced at lower strain rates. Specifically, while the percentage decrease is moderate at lower temperatures, it escalates significantly at higher temperatures. This indicates that as the material approaches its melt state, the impact of temperature on reducing its modulus becomes more evident and pronounced, with higher strain rates accelerating the reduction in mechanical stiffness. Consequently, higher thermal expansion and lower modulus occur above $T_{g}$ due to reduced intermolecular forces and enhanced molecular mobility. The alignment between the results derived from the application of the RJ model and those obtained through the MD simulation is notably evident. In this specific case, the Weibull modulus values are 4.3 and 6.2 for systems subjected to strain rates of 1.0 × 10^−5^ fs^−1^ and 1.0 × 10^−6^ fs^−1^, respectively, as demonstrated in Table 2. This concordance underscores the reliability and consistency of the RJ model in capturing the mechanical behavior of systems under varying strain rates. On the contrary, Figure 3b shows the opposite trend. With increasing temperature, the Poisson’s ratio starts to increase and is consistently higher for the system at the lower strain rate of 1.0 × 10^−6^ fs^−1^ throughout the entire temperature range, indicating greater ductility. At a strain rate of 1.0 × 10^−5^ fs^−1^, the ratio increases from 10.19% at 270 K to 77.82% at 400 K compared to its value at 150 K. At a strain rate of 1.0 × 10^−6^, the ratio increases from 17.66% at 270 K to 65.13% at 400 K. The increase in the Poisson’s ratio at higher strain rates is not abrupt, suggesting a limited lateral deformation of the material, indicating that it becomes stiffer at high strain rates. This contrasts with the behavior observed at lower strain rates. Furthermore, it is apparent how closely the results of the Poisson ratio derived from the RJ model align with those from MD simulations.

The analysis of Young’s modulus and Poisson’s ratio in relation to temperature, as presented in Figure 4, highlights the material’s thermal sensitivity and its transition from a rigid to a more flexible state as temperature increases. The marked decrease in Young’s modulus at temperatures above $T_{g}$ and at lower strain rates underscores the significant impact of thermal energy on the material’s stiffness. Additionally, the increase in the Poisson’s ratio at elevated temperatures indicates greater lateral deformation and enhanced ductility, consistent with the expected behavior of materials approaching their melting point.

When comparing the data in Figure 3 and Figure 4 on the Young’s modulus and Poisson’s ratio for the PEO/SiO_2_ nanocomposite system, it becomes clear that for the current PEO/SiO_2_ systems, there exists a “critical” strain rate value, approximately equal to 1.0 × 10^−5^ fs^−1^. Below this threshold, the system exhibits the characteristics of a rigid and brittle material, characterized by a high Young’s modulus and a low Poisson’s ratio. On the contrary, above this strain rate value, the system displays softer behavior, accompanied by an increase in the Poisson’s ratio and reduced rigidity (lower Young’s modulus). This behavior closely resembles the definition of the glass transition temperature ($T_{g}$). Below $T_{g}$, the system assumes a glassy state, demonstrating higher rigidity when compared to conditions at higher temperatures above $T_{g}$.

### 3.2. Heterogeneous (Local) Mechanical Behavior of PEO/SiO_2_ Nanocomposites

To investigate the heterogeneous mechanical characteristics exhibited by the PEO/SiO_2_ model systems, a thorough examination of the stress and strain fields at the atomic scale is imperative. To accomplish this, a per-atom assessment of stress and strain is employed, applied within the framework of a globally imposed strain. The computation of local stress at each individual atom, as mentioned above, is performed directly using the principles of the atomic virial formalism. Based on our previous research on polymer/silica nanocomposites [11], the determination of the extent of the interphase region surrounding the nanoparticle is based on data related to the density profile of the PEO chains with respect to their distance from the center of mass of the silica nanoparticle. As described in our previous works [11,72], it was found that the interphase region extends to a distance of 6.0 Å from the surface of the silica nanoparticle, coinciding with the initial dip observed in the density-radial distance curve. Following this, the matrix region is identified, beginning at the end of the interphase region. In this study, the matrix region is defined as the remaining polymers surrounding the interphase region (Figure 5). Based on the aforementioned statement, the calculation of stress within a particular domain, such as the PEO/SiO_2_ interface and the PEO matrix, involves the sum of individual contributions of each atom within the domain, followed by the division of this amount by the volume of the domain; the latter is due to the fact that the values of the atom array are in ‘pressure times volume’.

Figure 6 shows a three-dimensional (3D) visualization that illustrates the modulus of changes in elasticity in the interphase and matrix regions, revealing how they are influenced by temperature and strain rate. For better clarification, changes in the Young’s modulus for the interphase and matrix regions are analyzed based on the strain rate and temperature in Section 3.2.1 and Section 3.2.2, respectively.

#### 3.2.1. Dependence of the Local Linear Mechanical Properties on Strain Rate

To examine the behavior of the PEO/SiO_2_ nanocomposite system under varying strain rates, a comprehensive examination was carried out in three different temperatures chosen with respect to the average glass transition temperature, $T_{g}$, of the model PEO/SiO_2_ systems: (a) below (220 K), (b) equal to (270 K), and (c) above (330 K) the $T_{g}$. The resulting observations related to the elastic modulus in relation to the applied strain rate (graphed on a logarithmic scale) for these three distinct systems are visually presented in Figure 7. Furthermore, the graphs depicted the results derived from the RJ model, illustrating a strong similarity between the model results and those produced by the MD simulations, thus supporting our findings.

Based on the information provided in Figure 7a,b, when dealing with systems characterized by low temperature values and higher strain rates (exceeding 1.0 × 10^−5^ fs^−1^), the small deformation time and the glassy state result in both the interphase and matrix regions that display a similar level of rigidity. This observation can be attributed to the limited mobility of polymer chains in the glassy state, where thermal energy is insufficient to facilitate significant molecular rearrangements. Consequently, both the interphase and matrix are locked into a rigid structure, leading to uniform mechanical behavior across these regions. However, a more pronounced rigidity disparity between the interphase and matrix regions becomes evident if low strain rate values are applied. At these lower strain rates, there is sufficient time for the polymer chains in the matrix to partially relax, allowing for some degree of molecular mobility that is less present in the interphase, thereby creating a noticeable difference in rigidity. On the contrary, opposing results are observed for systems at higher temperature values (in a melt state, Figure 7c). In such systems, higher strain rate values lead to a greater rigidity gap between the interphase and matrix regions, attributed to the enhanced non-bonded interactions in the interphase of melty systems compared to the matrix region. Increased rigidity in the interphase can be attributed to the reorganization of molecular chains, which occurs at higher strain rates. At these elevated temperatures, when higher strain rates are applied, the nanoparticles exhibit a more pronounced ability to create a region with distinct properties compared to the matrix region farther from the nanoparticles. This results in a greater rigidity disparity between the interphase and the matrix, with the interphase becoming more structured and resistant, while the matrix remains less ordered and exhibits lower resistance. In melty systems at low strain rates, the Young’s modulus values for the interphase and matrix regions come closer together. This convergence stems from the fact that low strain rates and high temperatures magnify the decrease in rigidity in both the interphase and matrix regions. Under these conditions, the thermal energy allows for significant chain mobility in both regions, effectively reducing the overall rigidity and minimizing the differences between the interphase and matrix. It is evident from Table 2 that the Weibull modulus is higher for the matrix compared to the interphase region, with an increase as the system approaches a molten state.

#### 3.2.2. Dependence of the Local Linear Mechanical Properties on Temperature

According to Figure 8, a marked decrease in rigidity is observed in both the interphase and the matrix regions when the systems are exposed to temperatures that exceed the glassy transition temperature, indicating a transition to melt systems. As previously mentioned, the decrease in rigidity is due to increased molecular mobility and weakened intermolecular forces at elevated temperatures, resulting in the overall softening of both regions. On the contrary, comparable rigidity (E) is observed for the interphase and matrix regions at low temperature values, especially in systems subjected to high strain rates. At lower temperatures, the material maintains its rigidity due to the lower molecular mobility and stronger intermolecular interactions. High strain rates further exacerbate this rigidity by impeding the relaxation and reorientation of molecular chains, thus increasing material stiffness. The simultaneous presence of low strain rates and high temperatures not only accentuates the reduction in rigidity in both the interphase and matrix regions but also, at extremely high temperatures, results in similar rigidity for both regions. This phenomenon occurs because, when low strain rates coincide with exceedingly high temperatures, the concept of relevance of the ‘interphase region’ diminishes. In such extreme conditions, both the interphase and matrix regions undergo similar softening, making the previously observed differences less pronounced. Similar deductions can be drawn from the findings obtained through the RJ model, which closely match those arising from the MD simulations. This alignment reaffirms the enduring low stiffness observed within both the interphase and matrix sections of systems exposed to extreme temperatures and reduced strain rates, thus backing the notion of a diminishing interphase region under such conditions. Similar trends in the Weibull modulus as discussed earlier are apparent in Table 2, where the matrix exhibits a higher Weibull modulus compared to the interphase regions, with an increase observed as the strain rate decreases.

#### 3.2.3. Strain Heterogeneities within Polymer Nanocomposite Systems

Next, a detailed investigation is provided on how the local strain distribution evolves across different strain rates and temperatures within the PEO/SiO_2_ nanocomposite system, shedding light on the spatial variation and rate dependence of mechanical properties within the material. The heterogeneities of local stress and strain fields are examined for specific values of the applied global deformation, that is, strain values of 0.03, 0.06, and 0.09, each representing a specific frame within the elastic regime. The strain probability distribution function, P(ε), is computed for these strain values for both interphase and bulk regions. Data about the standard deviations of P(ε) for the interface and the matrix regions are presented in Figure 9, whereas in the inset, the entire P(ε) is shown. Systems at 220 K, 270 K, and 330 K are shown. The P(ε) data show the varying levels of strain heterogeneity that occur under different conditions, while the standard deviation curves offer information on the degree of variation in local strain within the material. It is clear that the local strain distributions are symmetrical, with initially narrow peaks signifying extensive adaptation to the global strain. As the strain increases, these distributions become more extensive, suggesting the presence of a more varied local strain field within the polymer. The significant fluctuations and variations observed in the standard deviation can be attributed to the heterogeneity present in the local strain distribution.

From the above results, it is clear that higher strain rates, such as 1.0 × 10^−4^ fs^−1^, produce narrower probability distribution curves, indicating a more uniform distribution of local strain (close to the global applied strain) within the material, whereas lower strain rates, such as 1.0 × 10^−7^ fs^−1^, produce wider probability distributions, indicating increased strain heterogeneities within the polymer nanocomposite. At the same time, lower temperatures, such as 220 K, result in narrower probability distributions, indicating a more homogeneous strain distribution. As temperature increases, the P(ε) becomes broader, indicating increased strain heterogeneity due to increased thermal fluctuations.

Furthermore, the bulk and interphase regions exhibit unique characteristics. In the bulk region, probability distributions are narrower, especially at lower temperatures and higher strain rates, indicating a more uniform strain distribution. However, in the interphase region, larger probability distributions are consistently detected under all conditions, indicating greater variability of the strain than in the bulk. Standard deviation curves support these findings, with lower strain rates and higher temperatures associated with higher standard deviation values, indicating greater variability in strain within the material. On the contrary, higher strain rates and lower temperatures produce smaller standard deviation values of P(ε), implying a more consistent strain distribution. These findings highlight the intricate interplay of strain rate, temperature, and the heterogeneity of local strain distribution in the PEO/SiO_2_ nanocomposite systems. They emphasize the need to take these aspects into account when assessing the mechanical behavior of such materials, especially in cases where the uniformity of the strain distribution is vital.

## 4. Mobility of Polymer Chains under Deformation

The investigation is extended to examine the motion of the polymer chains within the interphase- and bulk-like regions under various strain rates for systems at different temperature conditions: below, equal to, and above the glass transition temperature ($T_{g}$). The correlation between the mechanical response of the PEO/SiO_2_ systems and their mobility are explored by directly measuring the mean-squared displacement (MSD) of the polymer atoms under a specific deformation (strain, *ε*). The MSD, Δ*R*(ε(*t*)), is defined as $∆R(\varepsilon \left(t\right))=\langle \left({R\left(\varepsilon \left(t\right)\right)-R\left(\varepsilon \left(0\right)\right)}^{2}\right)\rangle$ and is analyzed across different temperatures. To focus on the non-affine displacements, the affine deformation $x_{0}^{m}\varepsilon_{x}$ is subtracted from the MSDs along the x direction (deformation), while $\nu y_{0}^{m}\varepsilon_{x}$ and $\nu z_{0}^{m}\varepsilon_{x}$ are subtracted from the MSDs in the y and z directions, respectively. Here, $x_{0}^{m}$, $y_{0}^{m}$, and $z_{0}^{m}$ represent the initial positions of atom m at equilibrium, and ν is the Poisson’s ratio at the current temperature. In this relation, *R*(ε(*t*)) and *R*(0) represent the positions of the atoms at time *t* and *t* = 0, respectively, with the brackets ⟨⟩ indicating a statistical average over all polymer atoms and all possible time origins.

Based on Figure 10a, the polymer mobility in the axial direction during uniaxial deformation is significantly influenced by the temperature of the system, particularly in relation to the glass transition temperature ($T_{g}$). For temperatures at or below $T_{g}$, where the polymer exists in a glassy state, the effect of varying strain rates on polymer motion in the interphase regions is minimal. This indicates that the polymer chains in these regions are relatively rigid, limiting their mobility regardless of the strain rate. However, when the temperature exceeds $T_{g}$, transitioning the polymer into a molten state, the impact of strain rate becomes more pronounced. In these higher temperature conditions, the matrix or bulk-like region exhibits a much larger disparity in MSD values at lower strain rates, particularly at 1.0 × 10^−7^ fs^−1^, compared to other strain rates. This suggests that at elevated temperatures and lower strain rates, the polymer chains in the matrix region have greater mobility, which could be due to the reduced viscosity and increased chain flexibility (entropic effect) in the molten state. Consequently, higher temperatures coupled with lower strain rates result in significantly increased MSD, with the matrix region showing a more pronounced response compared to the interphase region (enthalpic effect). This behavior highlights the complex interplay between temperature, strain rate, and polymer mobility, particularly across different regions of the polymer composite.

A similar trend is observed in Figure 10b when analyzing the total MSD values (sum of MSD values along x, y, and z directions). For systems at or below $T_{g}$, the total MSD is predominantly influenced by the x-direction, with negligible contributions from the y and z directions. This indicates that polymer motion in the glassy state is highly constrained and primarily occurs along the stretching axis. However, in the molten state, the MSD increases significantly, especially in the matrix or bulk-like region. This effect is particularly pronounced at lower strain rates (1.0 × 10^−7^ fs^−1^), where the reduced strain rate allows for greater molecular mobility, leading to much higher MSD values. This suggests that in the molten state, polymer chains have more freedom to move, and lower strain rates further facilitate this motion.

## 5. Conclusions

In this study, the mechanical behavior of PEO/SiO_2_ nanocomposites was explored using atomistic MD simulations, which is a powerful and efficient approach to studying intricate nanoscale interactions in nanocomposite systems, bridging the gap between atomistic details and macroscopic mechanical properties. Our investigation focusses on the effects of temperature, across the melt and glassy states, and the strain rate on the material’s response to external loads. Furthermore, the results obtained from the MD simulations are compared with those from the continuum RJ model. Through this comprehensive investigation, our objective is to shed light on the intriguing properties of PEO/SiO_2_ nanocomposites, paving the way for their enhanced application in various fields of engineering and nanotechnology. On the basis of the aforementioned explanations and discussions, the following key conclusions can be drawn:When exploring the influence of temperature on mechanical behavior, the study reveals that an increase in temperature leads to a reduction in material strength. For instance, the decrease in Young’s modulus increases from 6.69% to 64.79% as the system’s temperature rises from 200 K to 400 K. This drop in rigidity becomes more pronounced at lower strain rates.At temperatures below the glass transition temperature ($T_{g}$), nanocomposite systems tend to have a higher Young’s modulus, indicating a more rigid and glassy state. Above $T_{g}$, nanocomposites become softer and less rigid, resulting in a decrease in Young’s modulus.Strain rate variations have a more significant impact on the Young’s modulus at higher temperatures. The critical strain rate of approximately 1.0 × 10^−5^ fs^−1^ signifies a shift from brittle and rigid behavior to softer characteristics in the nanocomposite system.Strain rate increases lead to a considerable rise in Young’s modulus, with a peak increase of up to 99.7% when the strain rate is elevated from 1.0 × 10^−7^ fs^−1^ to 1.0 × 10^−4^ fs^−1^ at 220 K. This sensitivity to strain rate enhances with higher temperatures, indicating a transition toward improved strain resistance and deformation resistance.The transition from the glassy state to the molten one at temperature equal to $T_{g}$ has an equivalent impact on both the Young’s modulus and Poisson’s ratio. Nanocomposites with temperatures below $T_{g}$ may undergo less lateral expansion under deformation. They demonstrate greater Poisson’s ratios above $T_{g}$, increasing to 77.82% at 400 K for a strain rate of 1.0 × 10^−5^ fs^−1^ and 65.13% for 1.0 × 10^−6^ fs^−1^.An analysis of local results, which examine atomic-scale stress and strain fields, provides valuable information on the diverse mechanical properties exhibited by the PEO/SiO_2_ model systems. The discernment of interphase and matrix regions reveals significant variations in rigidity between these areas when subjected to varying strain rates and temperatures, thus aiding in the comprehension of local stress distribution.Differences between the (more rigid) interphase and the matrix region are more important for low temperatures and/or high strain rates. On the contrary, under conditions of extremely high temperatures and low strain rates, the differences in the mechanical behavior between the interphase and matrix regions are reduced, supporting the idea of a diminishing interphase zone.The precision of the Richeton–Ji (RJ) model in predicting the mechanical properties of the PEO/SiO_2_ nanocomposite system has been established, as its results are highly congruent with those obtained from MD simulations.The analysis reveals that temperature and strain rate significantly impact polymer chain mobility, especially above the glass transition temperature ($T_{g}$). At higher temperatures, lower strain rates result in greater mean-squared displacement (MSD) values, particularly in the matrix region. This underscores the importance of thermal and mechanical conditions in influencing polymer deformation, with the matrix region being more responsive than the interphase region.

In summary, this extensive investigation of deformed PEO/SiO_2_ systems, using atomistic MD simulations, has yielded profound insights into the intricate mechanical behavior influenced by the strain rate and temperature. The general analysis of the stress–strain measurements highlights the substantial impact of both the applied strain rate and the temperature conditions on the PEO/SiO_2_ systems. The single axial tensile test, used as a highly effective methodology, allows the precise determination of the Young’s modulus (E) and Poisson’s ratio (ν), pivotal indicators of material rigidity and deformation response.

## Figures and Tables

**Figure 1 polymers-16-02530-f001:**
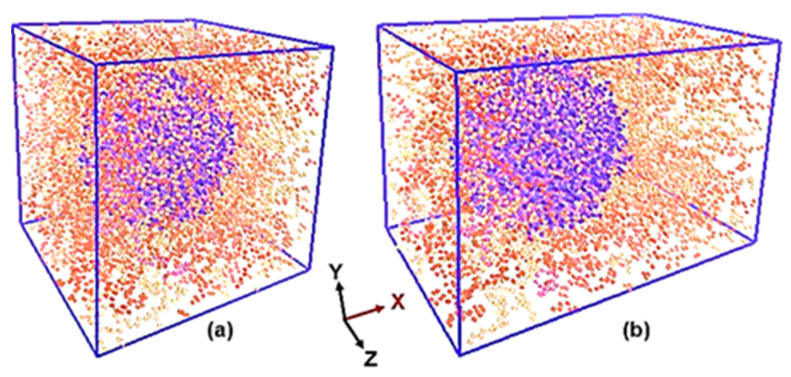
Snapshots of the PEO/SiO_2_ model systems, using periodically wrapped coordinates, in (**a**) initial (equilibrium) and (**b**) deformed in the x direction (ε = 0.4) configurations at T = 270 K. Blue dots represent the nearly spherical SiO_2_ nanoparticle, while red dots represent the surrounding PEO polymer in the unit cell system.

**Figure 2 polymers-16-02530-f002:**
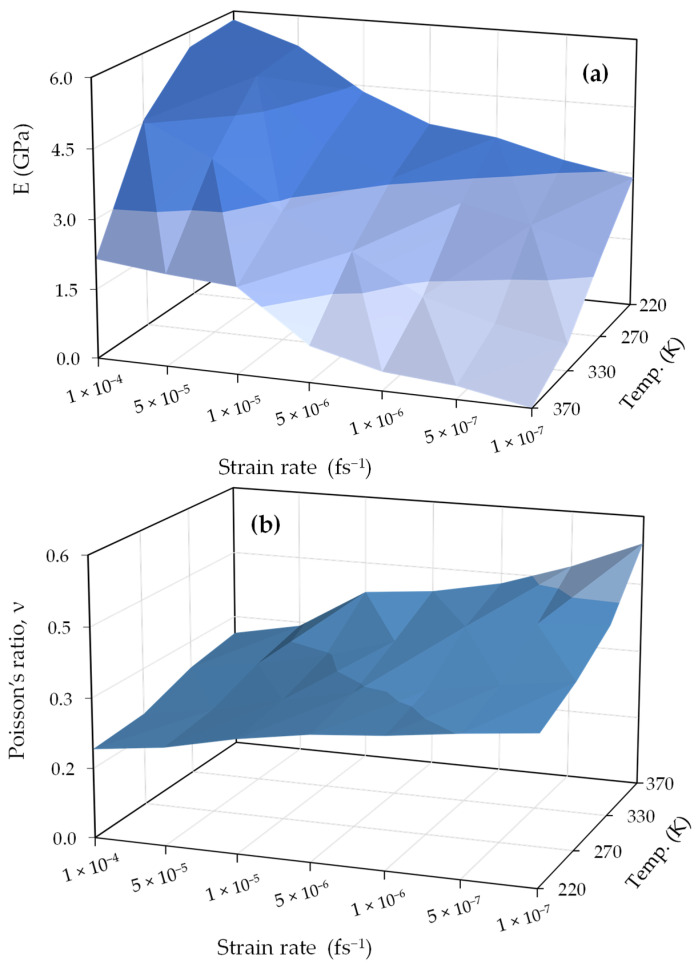
Variations in mechanical properties with temperature and strain rate: (**a**) Young modulus, (**b**) Poisson ratio. Darker colors indicate a transition towards a glassy state (higher E & lower ν values), while lighter colors represent a more fluid, melty state (lower E & higher ν values).

**Figure 3 polymers-16-02530-f003:**
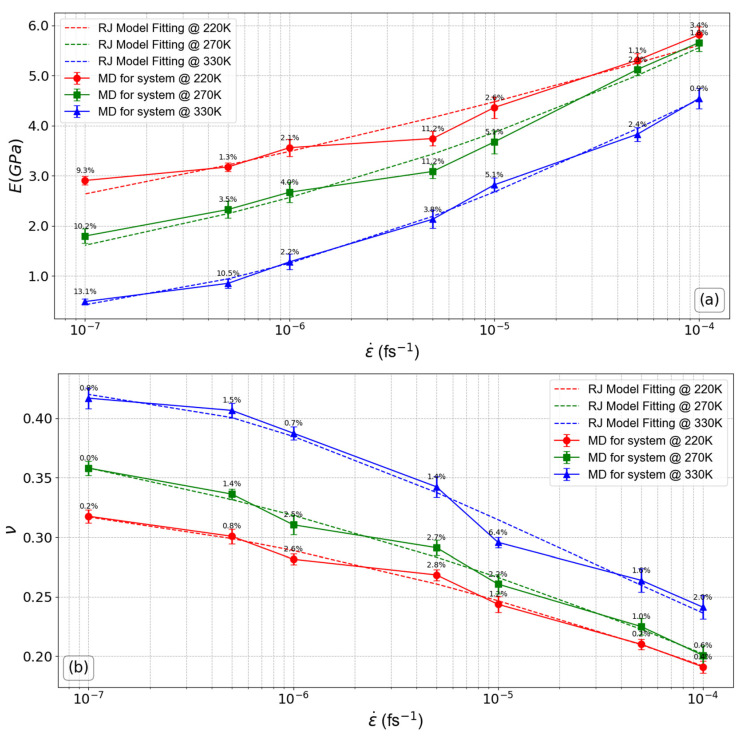
Linear mechanical properties of the PEO/SiO_2_ nanocomposites model as a function of strain rate: (**a**) Young’s modulus, (**b**) Poisson’s ratio. The error bars are computed by analyzing ten uncorrelated configurations. The percentage error values between the MD simulation results and the RJ model are displayed above each data point on the plot.

**Figure 4 polymers-16-02530-f004:**
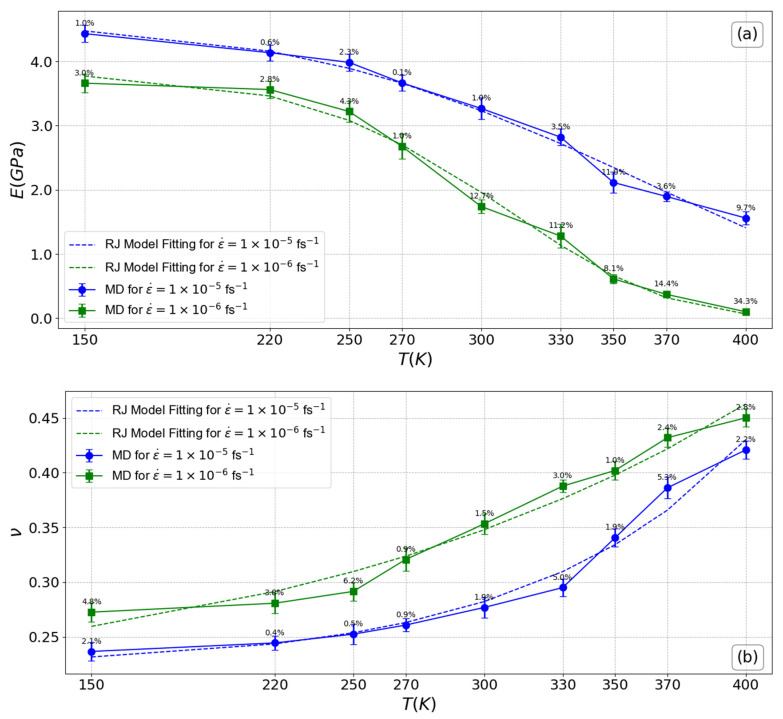
Linear mechanical properties of the PEO/SiO_2_ nanocomposite model as a function of temperature: (**a**) Young’s modulus, (**b**) Poison’s ratio. The error bars are computed by analyzing ten uncorrelated configurations. The percentage error values between the MD simulation results and the RJ model are displayed above each data point on the plot.

**Figure 5 polymers-16-02530-f005:**
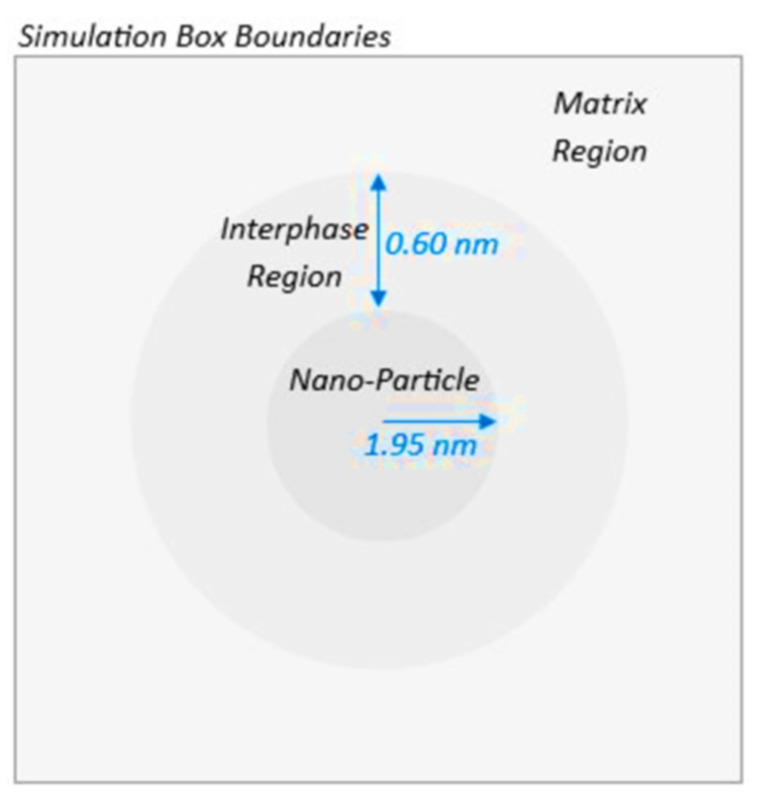
Schematic overview of designated zones within spheres around the nanoparticle in an in-plane box section.

**Figure 6 polymers-16-02530-f006:**
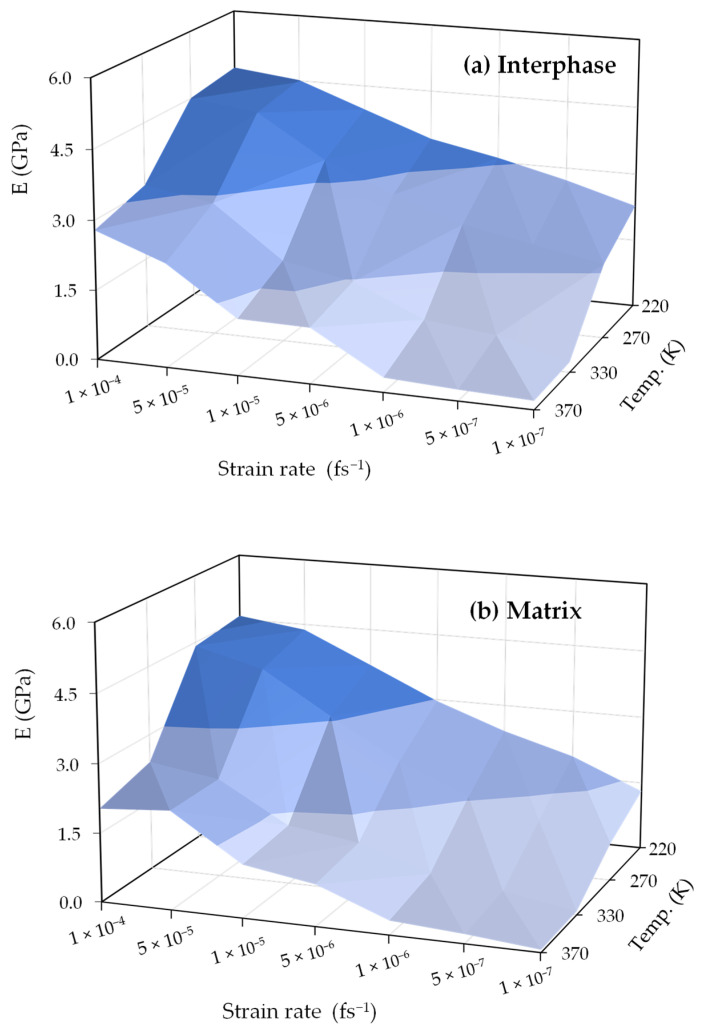
Elasticity modulus variations in (**a**) interphase and (**b**) matrix regions with temperature and strain rate dependency.

**Figure 7 polymers-16-02530-f007:**
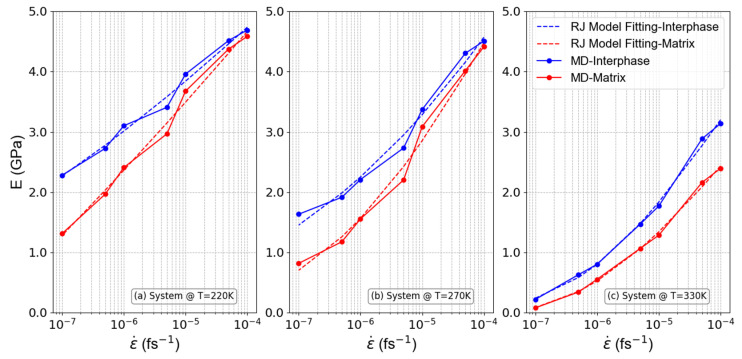
PNC’s elasticity modulus as a function of strain rate for systems at temperatures (**a**) below $T_{g}$ (220 K), (**b**) equal to $T_{g}$ (270 K), and (**c**) above $T_{g}$ (330 K).

**Figure 8 polymers-16-02530-f008:**
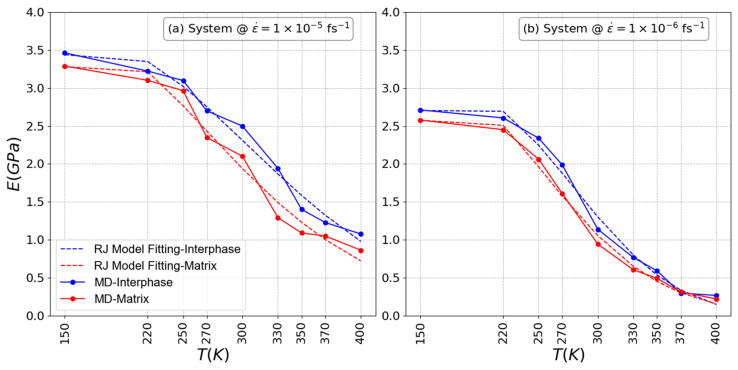
Elasticity modulus of PEO/SiO_2_ nanocomposites as a function of temperature for systems at strain rates (**a**) 1.0 × 10^−5^ fs^−1^ and (**b**) 1.0 × 10^−6^ fs^−1^.

**Figure 9 polymers-16-02530-f009:**
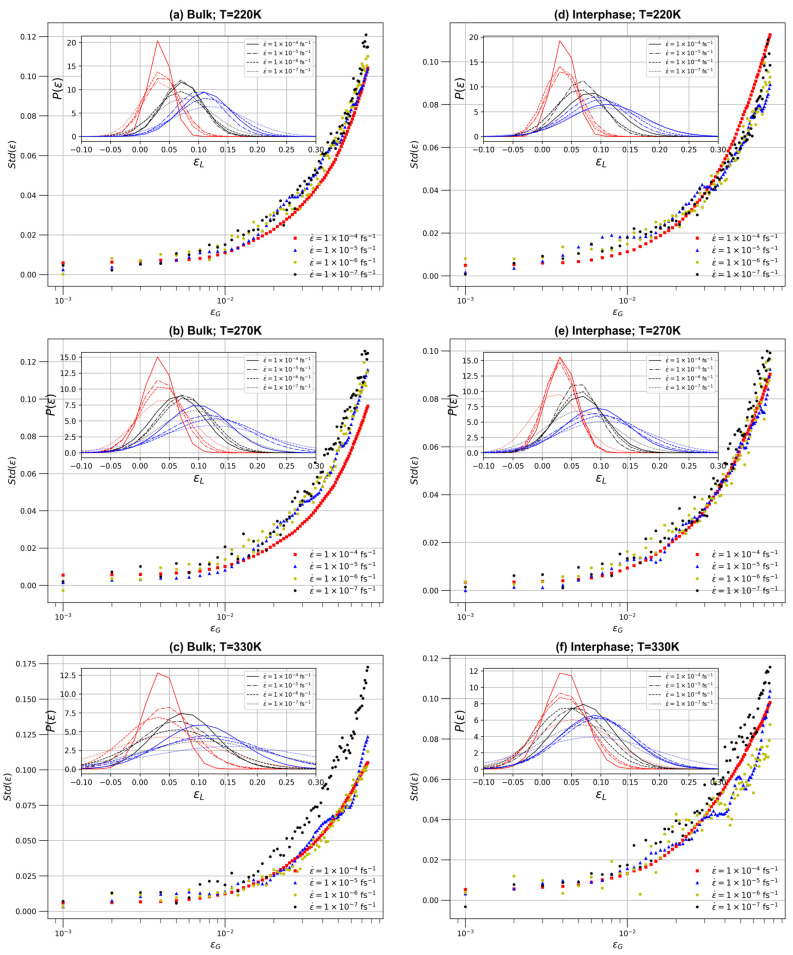
Analysis of the standard deviation with respect to the global strain and (inset) probability distribution of the local strain. The red, black, and blue curves represent the probability distribution in frames corresponding to global strains of 0.03, 0.06, and 0.09, respectively. The line styles for the red and blue inset plots match the legend of the black inset. (**a**–**c**) For bulk regions at 220 K, 270 K, and 330 K, respectively; and (**d**–**f**) for interphase regions at 220 K, 270 K, and 330 K, respectively.

**Figure 10 polymers-16-02530-f010:**
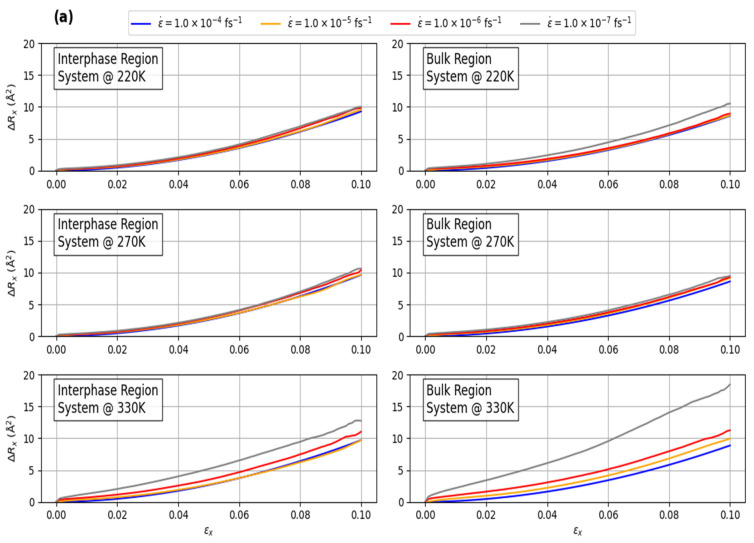
Variation in the mean-squared displacement (**a**) along x-direction and (**b**) for all components (sum of MSD along x, y, and z directions) in the interphase and matrix regions as a function of global applied strain under different strain rates, for systems at temperatures below (220 K), at (270 K), and above (330 K) the glass transition temperature ($T_{g}$).

**Table 1 polymers-16-02530-t001:** Temperature and strain rate values with the corresponding effective deformation time.

Temperature (K)	Strain Rate (fs^−1^)						
	1.0 × 10^−7^	5.0 × 10^−7^	1.0 × 10^−6^	5.0 × 10^−6^	1.0 × 10^−5^	5.0 × 10^−5^	1.0 × 10^−4^
	${t_{D}}^{1}(ps)$ = 1.0 × 10^4^	2.0 × 10^3^	1.0 × 10^3^	2.0 × 10^2^	1.0 × 10^2^	2.0 × 10^1^	1.0 × 10^1^
150	-	-	TE ^3^	-	TE	-	-
220	SRE ^2^	SRE	SRE/TE	SRE	SRE/TE	SRE	SRE
250	-	-	TE	-	TE	-	-
270	SRE	SRE	SRE/TE	SRE	SRE/TE	SRE	SRE
300	-	-	TE	-	TE	-	-
330	SRE	SRE	SRE/TE	SRE	SRE/TE	SRE	SRE
350	-	-	TE	-	TE	-	-
370	SRE	SRE	SRE/TE	SRE	SRE/TE	SRE	SRE
400	-	-	TE	-	TE	-	-

^1^ Corresponding effective deformation time for the strain rate values (in picoseconds). ^2^
The strain rate effect (SRE) is studied within a system that is kept at constant temperature. ^3^
The temperature effect (TE) is studied within a system under constant strain rate.

**Table 2 polymers-16-02530-t002:** Variations in the Weibull modulus of the RJ Model.

		Weibull Modulus (m)			
		Based on Young’s Modulus Formulation			Based on Poisson’s Ratio Formulation
		For Global Results	For Local Results		For Global Results
			Interphase	Matrix	
Strain rate effect	T = 220 K	1.80	1.47	2.02	5.90
	T = 270 K	2.26	2.47	6.35	2.47
	T = 330 K	7.40	9.95	16.70	−4.91
Temperature effect	$\dot{\varepsilon }=$ 1.0 × 10^−5^ fs^−1^	4.30	3.95	4.26	3.79
	$\dot{\varepsilon }=$ 1.0 × 10^−6^ fs^−1^	6.20	5.00	5.60	1.86

## Data Availability

The original contributions presented in the study are included in the article, further inquiries can be directed to the corresponding author.

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
