# Peer review of "Unraveling the Effect of Strain Rate and Temperature on the Heterogeneous Mechanical Behavior of Polymer Nanocomposites via Atomistic Simulations and Continuum Models"

_polymers, 2024, doi:10.3390/polym16172530_

Round 1

Reviewer 1 Report

Comments and Suggestions for Authors

The research has required to incorporate the suggestions/comments in the manuscript for better clarity in a few places to the researchers.

1. The author should remove I/we in the technical writing. Further, the abstract should be more quantitative in terms of improvement. Moreover, add a few lines on the possible application of obtained results in the abstract.

2. The author should elaborate on the research gap followed by the objective of the work. Further, what is the significance of citing too many references in one place such as [32,34,45,48,66,68,69].

3. The author should explain the assumptions taken in the MD simulation.

4. The author must add the discussion part of the obtained results with proper scientific explanation in Section 2.1.

5. The author should also mention the percentage error between MD simulation results and the RJ model.

6. What is the role of the interface in the MD simulation? Further, if the size of particles are changed then how it will affect the MD simulation?

Reviewer 2 Report

Comments and Suggestions for Authors

The authors conducted atomistic molecular dynamics simulations and continuum modeling to study the effect of temperature and strain rate on the mechanical properties of PEO/SiO2 nanocomposites. Specifically, they computed the changes in Youngs’s modulus and Poisson’s ratio with respect to changes in strain rate and temperature. Their findings revealed that an increase in strain rate causes an increase in Young’s modulus and a reduction in Poisson’s ratio. On the other hand, as temperature increases, Young’s modulus decreases, and Poisson’s ratio increases. They explain this behavior due to the ordered nature of polymer chains at higher strain rates, and the higher mobility caused by increased temperature. They also highlight the disparity between polymer behavior at the interface and in the bulk matrix. The authors report that higher strain rates and lower temperatures result in a homogenous strain distribution at the interface and in the bulk matrix. While the authors have used proper MD simulation techniques and adequate analysis methods to study the PEO/SiO2 nanocomposite system, a few issues need to be addressed before publication.

1)    In addition to bulk properties such as Youngs’s modulus and Poisson’s ratio, the authors should also focus on analyzing the local structural properties of polymer chains w.r.t temperature and strain rate to better explain the molecular origins of their observations. For example, the orientation distribution or the radius of gyration of the polymer chains at the interface and in bulk for different strain rates can help understand the effect of strain rate on polymer alignment and stretching. This is one of the main advantages of atomistic MD simulations, and the authors should utilize this opportunity to explore this further.

2)    From my understanding, the authors used all the data from MD simulations to fit the RJ model. So, what exactly is the model predicting? The authors should better highlight the purpose of the RJ model.

)    In Figure 1, the authors used unwrapped atom coordinates to capture the snapshot. Using periodically wrapped coordinates may help us better understand the simulation setup. 

Comments on the Quality of English Language

The authors should aim for more clarity and structure in explaining the temperature and strain-rate dependence of the mechanical properties. Specifically, the writing explaining the rigidity differences between the interface and the bulk matrix needs some attention.

Reviewer 3 Report

Comments and Suggestions for Authors

This research paper aims to model the effect of strain rate and temperature on the mechanical properties (Young's Modulus and Poisson's Ratio) for a PEO-SiO2 nanocomposite using MD simulations. The results from simulation are fitted to the Richeton-Ji (RJ) model. The manuscript is well structured and the results are presented in a logical manner. The design of the experiment is also clearly outlined and parameters used for the MD simulation are well documented. 

However, the RJ model used for fitting the simulation data seems to be the Richeton model for pure polymers and not the RJ model for polymer nanocomposites. The current model presented in the manuscript does not take into account the SiO2 nanofiller present in the polymer matrix. Hence, the authors should provide justification for why it is valid for fitting the results for the macroscopic behavior of the polymer nanocomposite. Alternatively, please use the RJ model (not the Richeton model) for fitting the data for the macro polymer nanocomposite. 

The Richeton model (used in the manuscript) could be used to fit the individual matrix and interphase polymers as done in section 3. Here, comparing the Weibull modulus determined for the matrix and interphase polymers vs neat polymer (from literature) will also help strengthen the case for the use of the Richeton model for the individual components and give greater insight to how the interphase is behaving vs the matrix under different conditions.

Some suggestions regarding the plots shown:  

1. Please consider condensing plots for Figure 3a and 3b  respectively from three to one each for Young's modulus and Poisson's ratio. Alternatively, if the plots are to be kept separate, please consider keeping the range of the y-axis for the similar plots to be the same. This will enable easier comparison of the data between each experimental condition and allow readers to make quicker comparisons. 

2. The same suggestion applies to Figures 4, 7 and 8. 

Comments on the Quality of English Language

Some minor corrections of language:

1. Line 269 says "... train Rate". It should be Strain Rate

2. The word brevity has been used twice in the manuscript. It seem that the word has been incorrectly used. Please revise. 

3. Line 245 should have a comma used- "... Young's modulus, whereas Poisson's ratio..."

Round 2

Reviewer 1 Report

Comments and Suggestions for Authors

No Comments

Author Response

Thank you for your note and for reviewing our manuscript. We appreciate your time and are glad that the revisions meet your expectations

Reviewer 2 Report

Comments and Suggestions for Authors

The authors adequately addressed all my comments. I think the manuscript is suitable for publication in its current form.

Author Response

Thank you very much for your positive feedback and for taking the time to review our manuscript. We are pleased to hear that you believe the manuscript is now suitable for publication. Your constructive comments have been invaluable in helping us refine and improve the quality of our work.

Reviewer 3 Report

Comments and Suggestions for Authors

Thank you to the authors for revising the manuscript. 

In the reply letter you have addressed that the model used for fitting the simulation data is the RJ model which takes into account the nanofillers. However, the equation 1 (line 298 of the manuscript) has no mention of the nanofiller parameters such as volume fraction of fillers or Elastic constant of fillers. Please see equations 15 and 16 for the RJ model in "A Review on the Modeling of the Elastic Modulus and Yield Stress of Polymers and Polymer Nanocomposites: Effect of Temperature, Loading Rate and Porosity" by R. Alasfar et. al.  These equations include nanofiller volume fraction, thickness of interphase, size of particles and the Young's modulus of nanoparticle.

Are these included in Eused in equation 1 of the manuscript? Based on the current presentation of equation 1, it reads that Ec is only dependent on the strain rate. If not, please add some additional details of how nanofiller parameters are incorporated in the model.

Additionally, the reference [32] used for equation 1 is "A unified model for stiffness modulus of amorphous polymers across transition temperatures and strain rates" which has data for neat polymers PMMA and PC. Please update this reference to the RJ model developed by Wang et. al.

The manuscript is otherwise well structured and reads well. The data is well presented and revisions made by the authors have enhanced the readability of the manuscript. Please add details for equation 1 to the manuscript to better explain the RJ model used. 

Round 3

Reviewer 3 Report

Comments and Suggestions for Authors

The addition of Appendix C has made the manuscript complete. Thank you for your revisions.